# SETD2 is required for DNA double-strand break repair and activation of the p53-mediated checkpoint

**Sílvia Carvalho[†], Alexandra C Vítor[†], Sreerama C Sridhara, Filipa B Martins, Ana C Raposo, Joana MP Desterro, João Ferreira, Sérgio F de Almeida\***

Instituto de Medicina Molecular, Faculdade de Medicina da Universidade de Lisboa, Lisboa, Portugal

**Abstract** Histone modifications establish the chromatin states that coordinate the DNA damage response. In this study, we show that SETD2, the enzyme that trimethylates histone H3 lysine 36 (H3K36me3), is required for ATM activation upon DNA double-strand breaks (DSBs). Moreover, we find that SETD2 is necessary for homologous recombination repair of DSBs by promoting the formation of RAD51 presynaptic filaments. In agreement, SETD2-mutant clear cell renal cell carcinoma (ccRCC) cells displayed impaired DNA damage signaling. However, despite the persistence of DNA lesions, SETD2-deficient cells failed to activate p53, a master guardian of the genome rarely mutated in ccRCC and showed decreased cell survival after DNA damage. We propose that this novel SETD2-dependent role provides a chromatin bookmarking instrument that facilitates signaling and repair of DSBs. In ccRCC, loss of SETD2 may afford an alternative mechanism for the inactivation of the p53-mediated checkpoint without the need for additional genetic mutations in TP53.

**\*For correspondence:**
sergioalmeida@medicina.
ulisboa.pt

[†]These authors contributed equally to this work

**Competing interests:** The authors declare that no competing interests exist.

**Reviewing editor**: Joaquin M Espinosa, Howard Hughes Medical Institute, University of Colorado, United States

## Introduction

DNA double-strand breaks (DSBs) are the most catastrophic form of DNA damage and pose great threat to genome stability. The major sensor of DSBs is ataxia telangiectasia mutated (ATM) kinase, which is critical for the initial steps of the DNA damage response (DDR). In response to DSBs, ATM phosphorylates and regulates the activity of several substrates involved in DNA repair, such as p53-binding protein 1 (53BP1) and histone H2AX (*Matsuoka et al., 2007*). To effectively repair a DSB, mammalian cells can choose from two different DDR pathways: nonhomologous end-joining (NHEJ) and homologous recombination (HR) (*Chapman et al., 2012*). During NHEJ, the broken DNA ends are blocked from 5′ end resection and held in close proximity by the Ku70-Ku80 heterodimer (Ku) (*Lieber, 2010*). HR is initiated when the 5′ DNA ends are resected by nucleases and helicases, generating two 3′ single-stranded DNA overhangs coated with phosphorylated replication protein A (RPA) that drives the formation of a RAD51 filament prior to strand invasion (*Chapman et al., 2012*). In contrast to NHEJ, which promotes direct ligation of the DSB ends in an error-prone manner and is available throughout the cell cycle, HR employs homologous sequences available after DNA replication as templates for error-free DNA repair (*San Filippo et al., 2008*). Both pathways proceed through a cascade of events whereby DNA damage sensors, transducers, and effectors detect and rejoin the broken DNA ends (*Harper and Elledge, 2007*). All these events take place within the relatively restricted environment of chromatin, the nucleoprotein complex of histones and DNA assembled into nucleosomes. DNA damage challenges chromatin integrity by eliciting the destabilization and reorganization of its structure (*Soria et al., 2012*). Conversely, a strictly regulated set of post-translational modifications of the histones N-terminal tails regulates the recruitment and activation of DDR factors

**eLife digest** Normal wear and tear, exposure to chemicals, and ultraviolet light can all damage DNA, so cells rely on a range of sensors and mechanisms to detect and repair damaged DNA. Cells also package DNA molecules inside structures called histones to protect them against damage.

Double-strand breaks—one of the most serious forms of DNA damage—are detected by an enzyme called ATM, and can be repaired in two ways. Bringing the broken strands back together is an obvious method, but it is also error prone. Using templates to generate new DNA to repair the damage is less prone to error, but it can only happen at certain times of the cell cycle.

Some cancers are linked to the faulty repair of double-strand breaks. Moreover, a type of kidney cancer called clear cell renal carcinoma is linked to a lack of activity by a protein called p53, even in individuals who don't have mutations in the gene for this protein. However, many people with this type of cancer have mutations in the gene for a protein called SETD2.

To investigate the links between SETD2 and DNA repair, Carvalho et al. compared cells with and without mutations in the gene for SETD2. It emerged that SETD2 must be present for DNA repair to take place: the SETD2 modifies the histones so that they can recruit the enzymes that repair the DNA via the template approach (which is relatively error free). SETD2 may be particularly important for repairing damage to genes without introducing errors.

Carvalho et al. also show that mutations in SETD2 are sufficient to inactivate p53. The gene for this protein, which impedes the proliferation of cells with genomic aberrations, such as double-strand breaks, is mutated in most cancers. Overall the results help to illustrate how histone modifications and the DNA damage repair mechanisms and checkpoints work in concert to suppress cancer.

(*Greenberg, 2011*). The most studied response of histones to DNA damage is the phosphorylation of the histone variant H2AX, which serves as a molecular beacon that signals the presence of DNA damage (*Rogakou et al., 1998*). Phosphorylated H2AX (γH2AX) arises within minutes after DNA damage and has a key role in the recruitment of multiple DDR factors to the repair centers (*Lukas et al., 2011*). In addition to γH2AX, other histone modifications have been shown to play key roles during the DDR, illustrating the importance of a thorough characterization of the chromatin landscape of a DNA lesion. We have recently shown that the histone methyltransferase SETD2 (also termed KMT3 or HYPB), which is responsible for all H3K36 trimethylation, but not H3K36 mono- or dimethylation (*Edmunds et al., 2008*), acts as a determinant of chromatin integrity by regulating nucleosome dynamics during transcription (*Carvalho et al., 2013*). A role of H3K36me3 in maintaining chromatin integrity was recently disclosed by the finding that the DNA mismatch repair (MMR) protein MutSα binds H3K36me3 and that SETD2 is necessary for human DNA MMR, a mechanism that corrects mismatches generated during DNA replication (*Li et al., 2013*). Nevertheless, whether SETD2 impinges on the cell's ability to cope with DNA DSBs is not yet known. In this study, we investigate the role of SETD2 on the cellular response to DNA DSBs. Our findings identify a novel role of this histone methyltransferase in rendering cells competent to repair DSBs. We demonstrate using powerful reporter assays that SETD2 is necessary for the DDR and that depletion of SETD2 impairs ATM activation and leads to delayed recruitment of 53BP1 to sites of DSBs. Our data further reveal that SETD2 is required for the recruitment of RAD51 to resected DNA 5′ ends and SETD2-depleted cells show reduced homology-directed repair of DSBs.

*SETD2* mutations were recently found in several cancers, such as clear cell renal cell carcinoma (ccRCC) (*Dalgliesh et al., 2010*; *Duns et al., 2010*; *Varela et al., 2011*; *Zhang et al., 2012*; *Fontebasso et al., 2013*; *Joseph et al., 2014*). In ccRCC, the p53-mediated cell cycle checkpoint is frequently inactivated despite the fact that the *TP53* tumor suppressor gene is rarely mutated (*Gurova et al., 2004*; *Dalgliesh et al., 2010*; *Sato et al., 2013*). This puzzling observation suggests that the p53 signaling in ccRCC might be repressed by an alternative mechanism. Herein, we further investigated whether the role of SETD2 in the DDR extends to the regulation of the p53-mediated checkpoint. We show that ccRCC cells carrying inactivating mutations on *SETD2* phenocopy the impaired DDR observed in *SETD2* RNAi-depleted human cells. Importantly, SETD2 inactivation severed the p53-dependent cell cycle checkpoint despite the persistence of unrepaired DNA lesions in ccRCC

cells. We propose that this unprecedented role of SETD2 in the DDR constitutes a novel tumor suppressor mechanism that could explain the high frequency of *SETD2* mutations found in several cancers and may provide an alternative mechanism for evasion of the p53-mediated checkpoint in *TP53* wt ccRCC cells.

## Results

### SETD2 is necessary for the recruitment and activation of early DDR factors

To assay how SETD2 impinges on the cellular response to chemically induced DSBs, we monitored the DDR by measuring the dynamics of phosphorylation of the major DSB sensor ATM. Human Osteosarcoma (U2OS) cells were challenged with three different DNA-damaging agents: the topoisomerase II inhibitor etoposide, which is known to induce a large amount of DSBs (*Burden et al., 1996*) and the radiomimetic dsDNA-cleaving agents neocarzinostatin (NCS) (*Goldberg, 1987*) and phleomycin (*Moore, 1988*). We depleted *SETD2* mRNA by RNA interference (RNAi) using three different synthetic small interfering RNA duplexes, which resulted in a global loss of the H3K36me3 histone mark that persisted throughout the entire chase periods following the DNA damage (*Figure 1A–C*). As a control, we used the GL2 duplex, which targets firefly luciferase (*Elbashir et al., 2001*). In control cells, the levels of H3K36me3 remained constant during the DDR and were undistinguishable from those of undamaged cells, suggesting that this histone mark is not amplified following the DSBs (*Figure 1A–C*). Analysis of the phosphorylation levels of ATM revealed that the DDR was promptly activated upon induction of DSBs with the three compounds (*Figure 1*). ATM phosphorylation (pATM) peaked at the early time points after each treatment in control cells (*Figure 1A–C*). In contrast, SETD2-depleted cells revealed a significant impairment in DNA damage signaling as revealed by decreased pATM levels detected upon treatment with each of the three drugs (*Figure 1A–C*). In agreement with impaired ATM activation, the phosphorylation levels of its downstream substrates H2AX and 53BP1 decreased in SETD2-depleted cells following treatment with NCS or, more appreciably, etoposide (*Figure 1A,B*). In DSBs induced by phleomycin, depletion of SETD2 had only a very mild impact on phosphorylation of 53BP1 or H2AX (*Figure 1C*) suggesting that either the remaining pATM is sufficient to transduce the DNA damage signaling or that alternative ATM-independent pathways operate in phleomycin-induced DSBs.

To directly visualize how does ablation of SETD2 impinge on 53BP1 nucleation at sites of DNA damage, we tracked 53BP1-GFP fusion proteins in live-cells upon induction of DSBs with a 405 nm

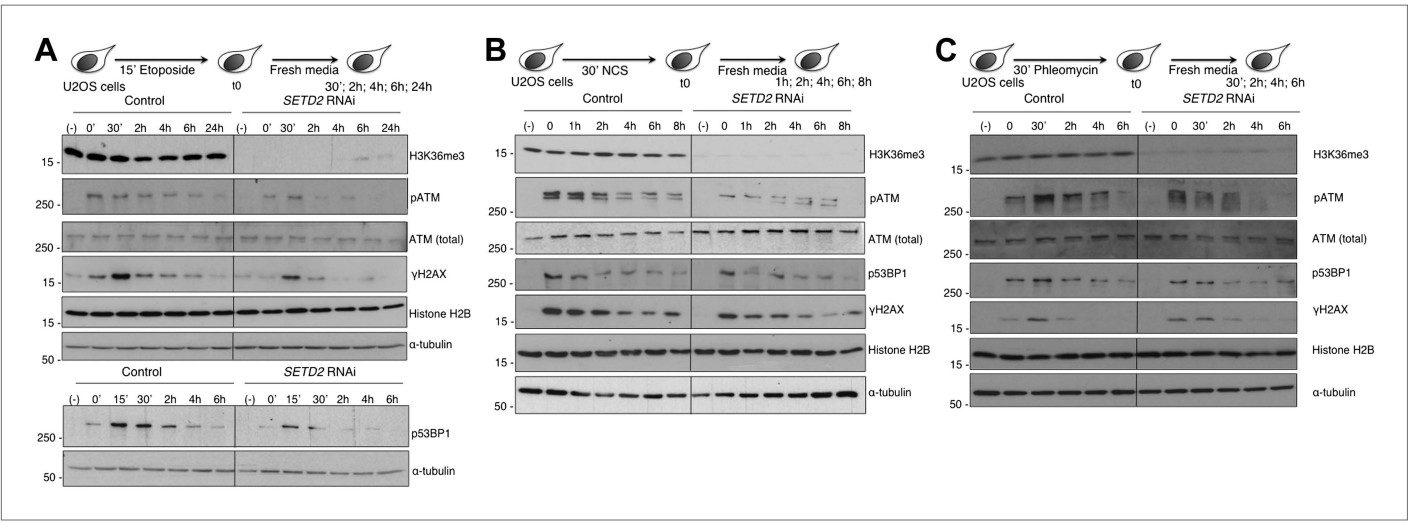

**Figure 1**. SETD2 is necessary for ATM activation during the DNA damage response. Control and *SETD2* RNAi-depleted U2OS cells were challenged with etoposide (**A**), NCS (**B**) or phleomycin (**C**) and chased in fresh media during the indicated time points. Western blot analysis was performed with antibodies against the indicated proteins. Molecular weight markers (KDa) are shown on the left of each blot. Data are from one representative experiment of at least three independent experiments performed with similar results.

laser (*Figure 2A*). In control cells, 53BP1-GFP was recruited to damaged chromatin within 2 min after laser micro-irradiation and was retained at the sites of damage during the 15 min of live-cell recording. In contrast, recruitment of 53BP1-GFP to irradiated chromatin was significantly delayed in SETD2-depleted cells (*Figure 2A*). Importantly, *SETD2* RNAi had no appreciable effects on the total cellular levels of 53BP1-GFP (*Figure 2B*).

We then investigated whether SETD2 is actively recruited to DSB sites. For that, we engineered a SETD2-GFP fusion protein and tracked it in live cells upon laser-induced DNA damage (*Figure 2C,D*). In agreement with the lack of fluctuations of the H3K36me3 levels following DSBs (*Figure 1A*), SETD2-GFP did not accumulate at the laser micro-irradiated regions (*Figure 2D*).

Since we did not find any evidence for de novo histone H3K36 trimethylation during the DDR, we inspected if H3K36me3 is present at the nucleosomes adjacent to the sites of DSBs. In order to have single-nucleosome resolution, U2OS nuclei extracts were digested with micrococcal nuclease (MNase). Analysis of the MNase-digested extracts revealed that the majority of the DNA fragments were approximately 150 bp, which is consistent with the length of mono-nucleosomal DNA (*Figure 2—figure supplement 1*). Furthermore, each of the four nucleosomal histones was detected in H3K36me3 pull downs, indicating that isolated nucleosomes remained intact (*Figure 2E*). 53BP1 and H3K36me3 were immunopurified from the mono-nucleosome preparations, and the complexes were immunoblotted with antibodies against γH2AX and H3K36me3 (*Figure 2F*). In control (i.e., undamaged) cells, γH2AX was neither detected in the input sample (total cell lysates) nor in any of the two pull-downs (*Figure 2F*). Upon treatment with NCS a robust signal for γH2AX was observed in the input lane and in the samples immunopurified with antibodies against 53BP1 or H3K36me3, indicating the activation of the DDR (*Figure 2F*). These data reveal that both 53BP1 and H3K36me3 are present in the same nucleosomes that acquire γH2AX upon DNA damage. Moreover, 53BP1 was detected in complexes immunoprecipitated with the H3K36me3 antibody upon induction of DSBs (*Figure 2F*). This suggests that H3K36me3 decorates at least a fraction of the nucleosomes to which 53BP1 binds. Altogether, these findings further support the view that H3K36me3 contributes to the establishment of the chromatin state that coordinates the recruitment and activation of DDR factors, even though SETD2 is not actively recruited to DNA damage sites.

## SETD2 is required for RAD51 recruitment during homologous recombination repair

The binding of 53BP1 to damaged chromatin tilts the balance between the DNA repair pathways towards NHEJ by preventing 5′ end resection of the broken DNA ends (*Bouwman et al., 2010*; *Bunting et al., 2010*; *Chapman et al., 2012*). We therefore investigated whether SETD2 depletion favors DNA 5′ end resection and repair of DSBs by HR. To this end, we quantified the DNA in the vicinity of a DSB created by the *I-SceI* nuclease using the DR-GFP reporter assay (*Pierce et al., 1999*; *Moynahan and Jasin, 2010*). Upon depletion of *SETD2* by RNAi, the amount of DNA measured by quantitative real-time PCR at 250 bp, 300 bp, 350 bp and 2200 bp apart from the *I-SceI* site was only 53%, 62%, 57%, and 71%, respectively, of the DNA levels present in control cells (*Figure 3A*). These data reveal that ablation of SETD2 leads to persistent DNA 5′ end resection of DSBs. In agreement, phosphorylation of RPA (pRPA), an event that is directly linked to efficient DNA 5′ end resection (*Sartori et al., 2007*), was detected in SETD2-depleted cells within the first hour after DNA damage with NCS and reached equilibrium after 6 to 8 hr, when its levels were similar to control cells (*Figure 3B*). Following DNA damage, the recruitment of RPA to DNA repair centers is revealed by the formation of repair-associated foci, a process that can be visualized by immunofluorescence (*Figure 3C*). At 8 hr after NCS treatment, widespread RPA foci were observed in the nuclei of both control and *SETD2* RNAi cells, suggesting that recruitment of RPA to DNA lesion sites is independent of SETD2 activity. As expected, RPA foci were visible only in cyclin A-positive cells that are in S or G2 phase of the cell cycle (*Figure 3C*).

The HR-mediated repair of DSBs requires the recruitment of different DNA-repair factors including BRCA1 to the site of damage. BRCA1 cooperates with Mre11–Rad50–Nbs1 (MRN) complex and C-terminal binding protein interacting protein (CtIP) to catalyze 5′ to 3′ DNA resection of the broken DNA ends (*Chapman et al., 2012*). Upon NCS treatment, abundant BRCA1 foci were formed in control and SETD2-depleted cells that stained positively for cyclin A (*Figure 4A*). This result is consistent with the SETD2-independent DNA 5′ end resection and suggests that BRCA1 recruitment to DSBs is largely unaffected by SETD2 depletion (*Figure 4A*).

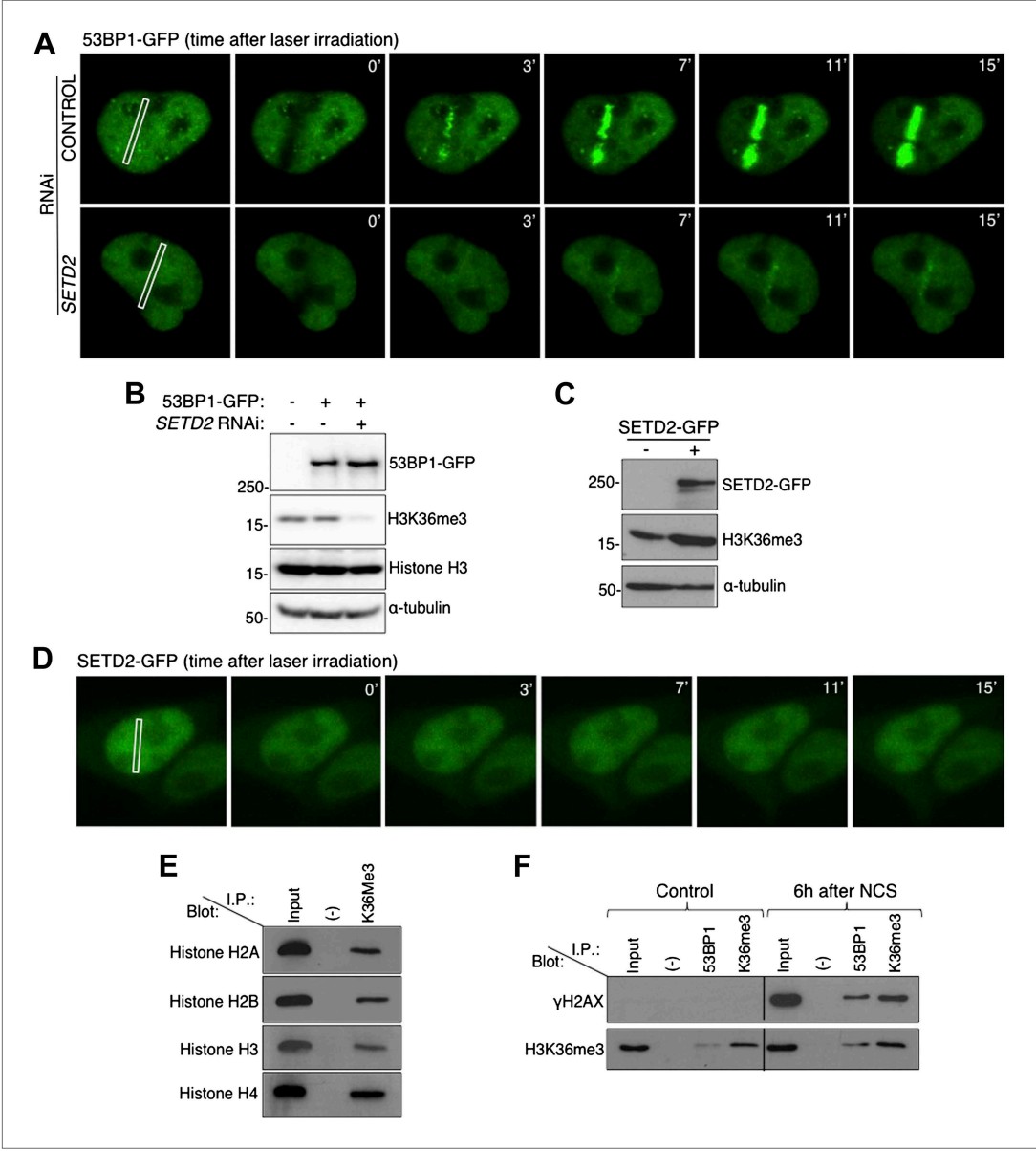

**Figure 2**. SETD2 promotes 53BP1 recruitment to DNA damage sites. (**A**) 53BP1-GFP transfected U2OS cells were damaged by laser irradiation of the indicated nuclear region. The dynamics of 53BP1-GFP during the DNA damage response on control and SETD2-depleted cells was monitored by live cell imaging for 15 min after laser irradiation. One representative experiment from over 50 individual cells recorded is shown. (**B**) Control and *SETD2* RNAi-depleted U2OS cells were transfected with a 53BP1-GFP expression plasmid. Western blot analysis was performed with antibodies against GFP (to detect 53BP1-GFP), H3K36me3, total histone H3, and α-tubulin. Molecular weight markers (KDa) are shown on the left. (**C**) SETD2-GFP transfected U2OS cells were lysed and processed for western blot with antibodies against GFP (to reveal SETD2-GFP), H3K36me3 and α-tubulin. Molecular weight markers (KDa) are shown on the left. (**D**) Live-cell images of SETD2-GFP dynamics were recorded upon laser-induced DNA damage of the indicated nuclear region of U2OS cells during 15 min (**E**) H3K36me3 was immunoprecipitated from MNase-digested U2OS cell extracts and each of the four nucleosomal histones was detected by immunoblotting of the SDS-PAGE resolved complexes. (**F**) Nuclear co-immunoprecipitations before and 6 hr after NCS treatment were performed on U2OS cells using the indicated antibodies (I.P.: 53BP1 and H3K36me3). The Input lane represents total cell lysates and (−) denotes the negative control immunoprecipitation (beads only). Purified complexes were resolved by SDS-PAGE and blotted with antibodies against γH2AX or H3K36me3.

The following figure supplements are available for figure 2:

**Figure supplement 1**. Preparation of single-nucleosome extracts for the co-immunoprecipitation experiments.

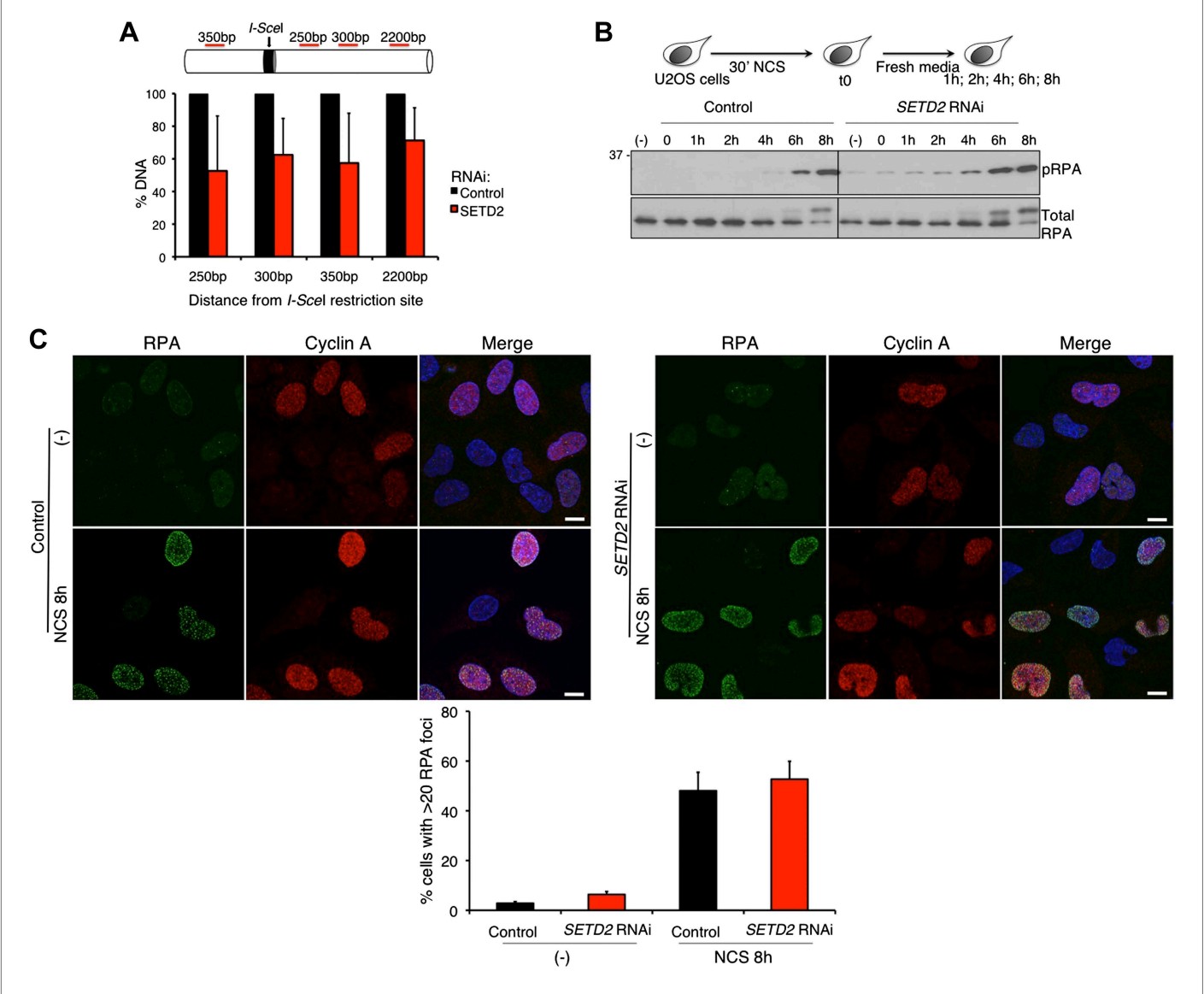

**Figure 3**. SETD2-independent DNA 5′ end resection and RPA recruitment to DSBs. (**A**) Direct measurement of DNA 5′ end resection at the *I-Sce*I site of the *GFP* gene in control and SETD2-depleted U2OS DR-GFP cells. The percentage of non-resected DNA at 250 bp, 300 bp, 350 bp, and 2200 bp from the *I-Sce*I restriction site is shown. The percentage of DNA at each location was set to 100% in control cells. Means and standard deviations from five independent experiments using two distinct siRNAs targeting *SETD2* are shown. (**B**) Control and *SETD2* RNAi-depleted U2OS cells were challenged with NCS during 30 min and chased in fresh media during the indicated time points. Western blot analysis was performed with antibodies against pRPA and total RPA. Molecular weight markers (KDa) are shown on the left. (**C**) Immunofluorescence of control and *SETD2* RNAi depleted U2OS cells before and 8 hr after NCS treatment was performed with antibodies against RPA and cyclin A. Dapi was added to the mounting medium to stain the DNA. Scale bars: 5 μm. The graph shows the percentage of cells with more than 20 RPA foci. A minimum of 400 cells on each of three independent experiments was scored.

BRCA1-mediated DNA 5′ end resection and phosphorylation of RPA precede the formation of RAD51 filaments on the DNA single-stranded 3′ overhangs that promote the search for homologous DNA sequences and strand exchange (***Wyman and Kanaar, 2006***). We then inspected the recruitment of RAD51 to broken DNA ends on control and SETD2-depleted cells. RAD51 foci were detected on cells that were in S or G2 phase, as revealed by the positive staining of cyclin A (***Figure 4B***). Notably, RAD51 was recruited to DSBs in a SETD2-dependent manner. Upon *SETD2* RNAi-depletion, the percentage of U2OS cells with ≥5 foci at 8 hr and 24 hr post-NCS treatment (17% and 30%, respectively) was significantly reduced in comparison to control cells (36% and 46%) (***Figure 4B***). These data establish a presynaptic role of SETD2 that promotes the recruitment of RAD51 to resected DNA ends.

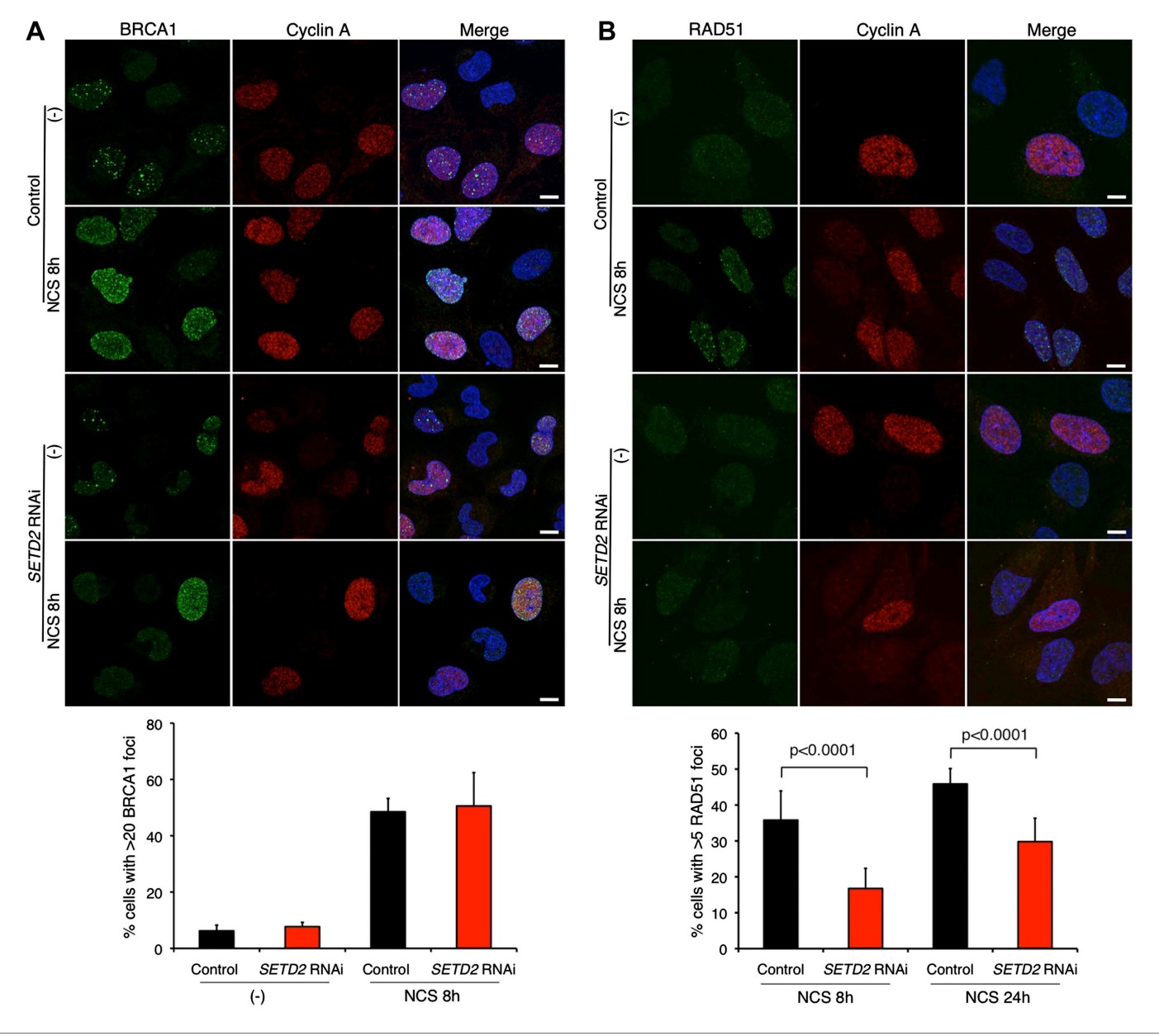

**Figure 4**. SETD2 promotes RAD51 recruitment to DSBs during DNA repair. Immunofluorescence of control and *SETD2* RNAi-depleted U2OS cells before and 8 hr after NCS treatment was performed with antibodies against BRCA1 (**A**) or RAD51 (**B**). Cyclin A staining was performed in parallel. Dapi was added to the mounting medium to stain the DNA. Scale bars: 5 µm. The graphs represent the average and standard deviations of the percentage of cells with BRCA1 or RAD51 foci after NCS treatment. A minimum of 500 cells from each of three individual experiments was scored for each experimental condition.

To directly evaluate the impact of this SETD2-dependent role on the efficiency of HR, we depleted SETD2 from U2OS cells stably expressing the DR-GFP reporter. The DR-GFP reporter allows homology-directed repair to be scored by flow cytometry owing to the correction of an inactive *GFP* gene through HR-dependent gene conversion (*Figure 5*) (*Pierce et al., 1999*; *Moynahan and Jasin, 2010*). Ectopic expression of *I-Sce*I in DR-GFP reporter cells led to an average cutting efficiency of the *I-Sce*I restriction site located on the inactive *GFP* gene of 35% (control cells) and 33% (*SETD2* RNAi). In agreement with the impaired RAD51 recruitment, SETD2-depleted cells exhibited a strong reduction (~50%) in HR without detectable differences in the cell cycle progression (*Figure 5A,B*).

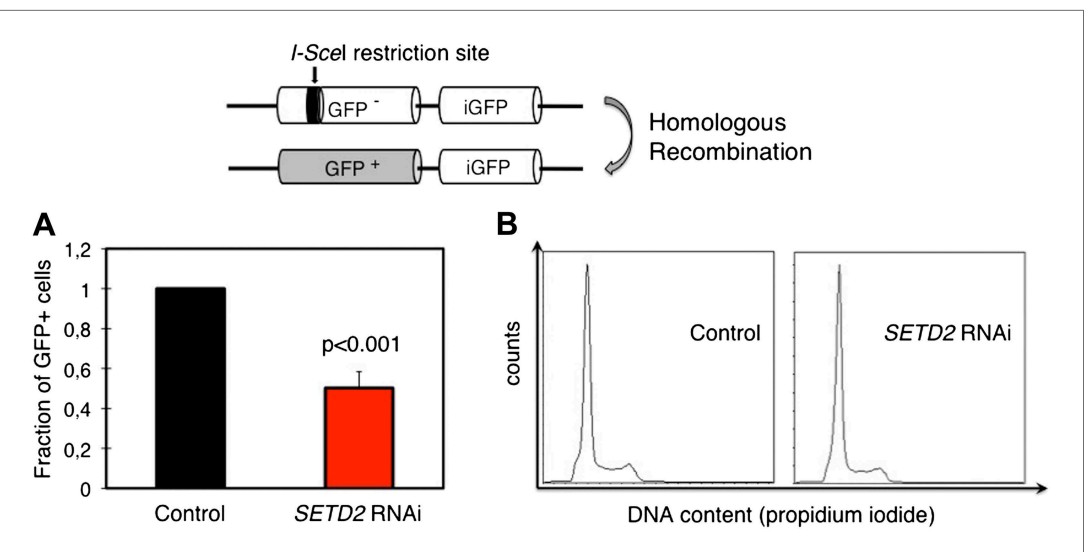

**Figure 5**. SETD2 is required for homologous recombination DNA repair. (**A**) The efficiency of homologous recombination in control and *SETD2* RNAi-depleted cells was investigated by measuring the fraction of GFP + cells in the U2OS DR-GFP reporter assay. Means and standard deviations from five independent experiments using two distinct siRNAs targeting *SETD2* are shown. (**B**) Cell cycle progression of control and SETD2-depleted U2OS DR-GFP cells was obtained by flow cytometry analysis of propidium iodide staining.

## ccRCC cells with *SETD2* inactivating mutations display impaired DDR

A high frequency of *SETD2* mutations was recently disclosed in several cancers, of which clear cell renal cell carcinoma (ccRCC) shows the highest *SETD2* mutation rate (***Dalgliesh et al., 2010***; ***Varela et al., 2011***). Our present findings suggest that impaired DDR proficiency constitutes a hallmark of *SETD2* mutant cancers and could reveal a novel mechanism through which SETD2 suppresses tumor development and growth. To further test this view, we inspected the DDR in ccRCC cell lines containing inactivating mutations in *SETD2* that result in a strong reduction of H3K36me3 levels (***Figure 6A***) (***Duns et al., 2010***). Notably, induction of DSBs in the ccRCC cells resulted in activation of the DDR in a SETD2-dependent manner. Two different *SETD2* wild-type (wt) cell lines (RCC-JW and Caki-2) responded to DNA damage with the rapid formation of γH2AX foci and ATM phosphorylation within the first 30 min of the DDR (***Figure 6B–D***). In contrast, DNA repair was significantly impaired in two *SETD2* mutant ccRCC cell lines (RCC-MF and RCC-FG2) as revealed by the lack of γH2AX foci and ATM phosphorylation upon induction of DSBs (***Figure 6B–D***). These data recapitulate the phenotype of *SETD2* RNAi-depleted U2OS cells and suggest that defects in the DDR are a hallmark of cancer cells with *SETD2* mutations. We then attempted to perform a genetic complementation experiment with our SETD2-GFP transgene to rescue the *SETD2* mutant phenotype. However, despite our efforts, we were not able to obtain a *SETD2* mutant ccRCC cell population expressing the transgene. One possibility is that loss of SETD2 function in the mutant ccRCC cells feeds a cellular proliferation mechanism that if disrupted upon transfection of the wt SETD2-GFP decreases cell viability. Nevertheless, we consistently observed a similar phenotype of impaired DDR in 4 different ccRCC cell lines (RCC-ER; -FG2; -AB; -MF) carrying inactivating mutations in distinct regions of *SETD2* (***Duns et al., 2010***). Amongst the *SETD2* mutant cell lines, RCC-FG2 had the lowest levels of H3K36me3 and showed the strongest impairment in the DDR signaling as revealed by the lack of detectable pATM and γH2AX (***Figure 6D***). We shall note that RCC-JW cells carry a missense mutation on *SETD2* exon 3, which does not interfere with the methyltransferase activity of the enzyme, and for this reason, we classified them as *SETD2* wt (***Figure 6C***) (***Duns et al., 2010***). In fact, concerning the DDR, RCC-JW cells behaved equally to two additional *SETD2* wt ccRCC cell lines (Caki-1 and Caki-2) that were also used on our experiments.

To investigate if alternative DDR pathways were able to efficiently repair the DSBs in the *SETD2* mutant cells, we measured the kinetics of DNA repair in ccRCC cells using comet assays (***Figure 7***).

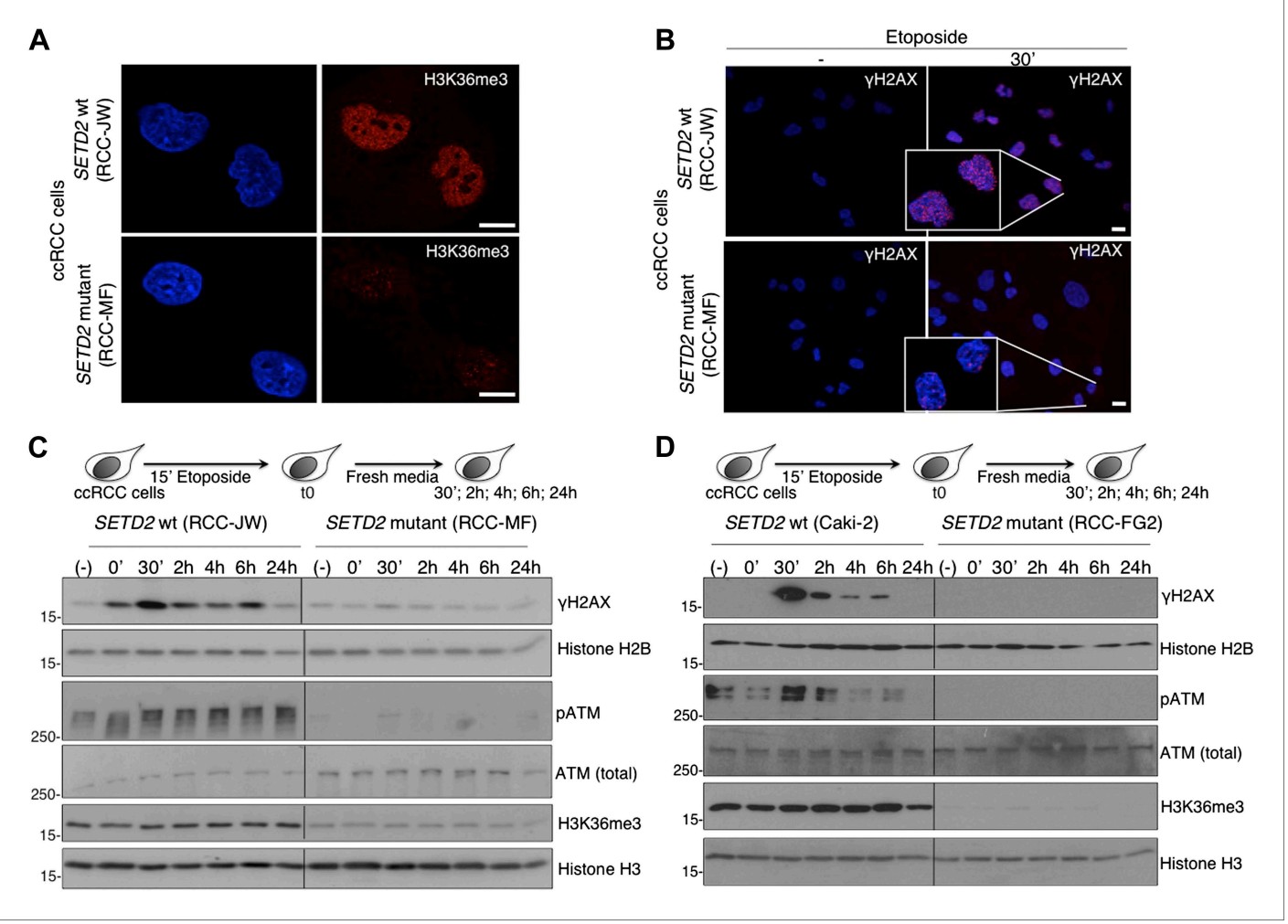

**Figure 6**. ccRCC cells with inactivating mutations on *SETD2* show impaired DNA damage signalling. The H3K36me3 levels of *SETD2* wt and mutant ccRCC cells were inspected with an antibody against H3K36me3 by immunofluorescence (**A**). (**B**) Immunofluorescence of *SETD2* wt and mutant ccRCC cells before and 30 min after etoposide treatment for 15 min was performed with an antibody against γH2AX. Dapi was added to the mounting medium to stain the DNA. Scale bars: 5 µm. (**C** and **D**) Two *SETD2* wt (RCC-JW and Caki-2) and two *SETD2* mutant (RCC-MF and RCC-FG2) cell lines were challenged with etoposide during 15 min and chased in fresh media during the indicated time points. Western blot analysis was performed with antibodies against the indicated proteins. Molecular weight markers (KDa) are shown on the left.

Chemical induction of DSBs yielded a similar degree of DNA damage in both *SETD2* wt and mutant ccRCC cells, as revealed by the comparable fraction of DNA that migrated away from the nuclei and was detected in the comets tails immediately after etoposide treatment (*Figure 7*). However, these cells differed in their ability to repair the chemically induced DSBs. The amount of fragmented DNA decreased 35% after 3 hr and 45% after 6 hr in the wt cells, which compared to a maximum of 27% reduction observed in mutant cells 6 hr after DNA damage (*Figure 7*). These results were recapitulated with an additional set of *SETD2* wt and mutant ccRCC cells, which repaired 49% (Caki-2) and 31% (RCC-FG2) of the DNA damage in 6 hr (*Figure 7—figure supplement 1*). These findings reveal that the impaired DDR observed in *SETD2* mutant cells translates into a massive failure to repair DSBs, which is likely to impact severely on genomic stability and may sustain the tumorigenic phenotype.

## Inactivation of SETD2 severs the p53-mediated checkpoint despite persistent DNA damage

In circumstances where DNA damage reaches an irreversible stage, specific cell cycle checkpoints are in place to prevent catastrophic cell cycle progression (*Kastan and Bartek, 2004*; *Sperka et al., 2012*). The DNA damage checkpoints employ sensor proteins, such as ATM, to detect DNA damage

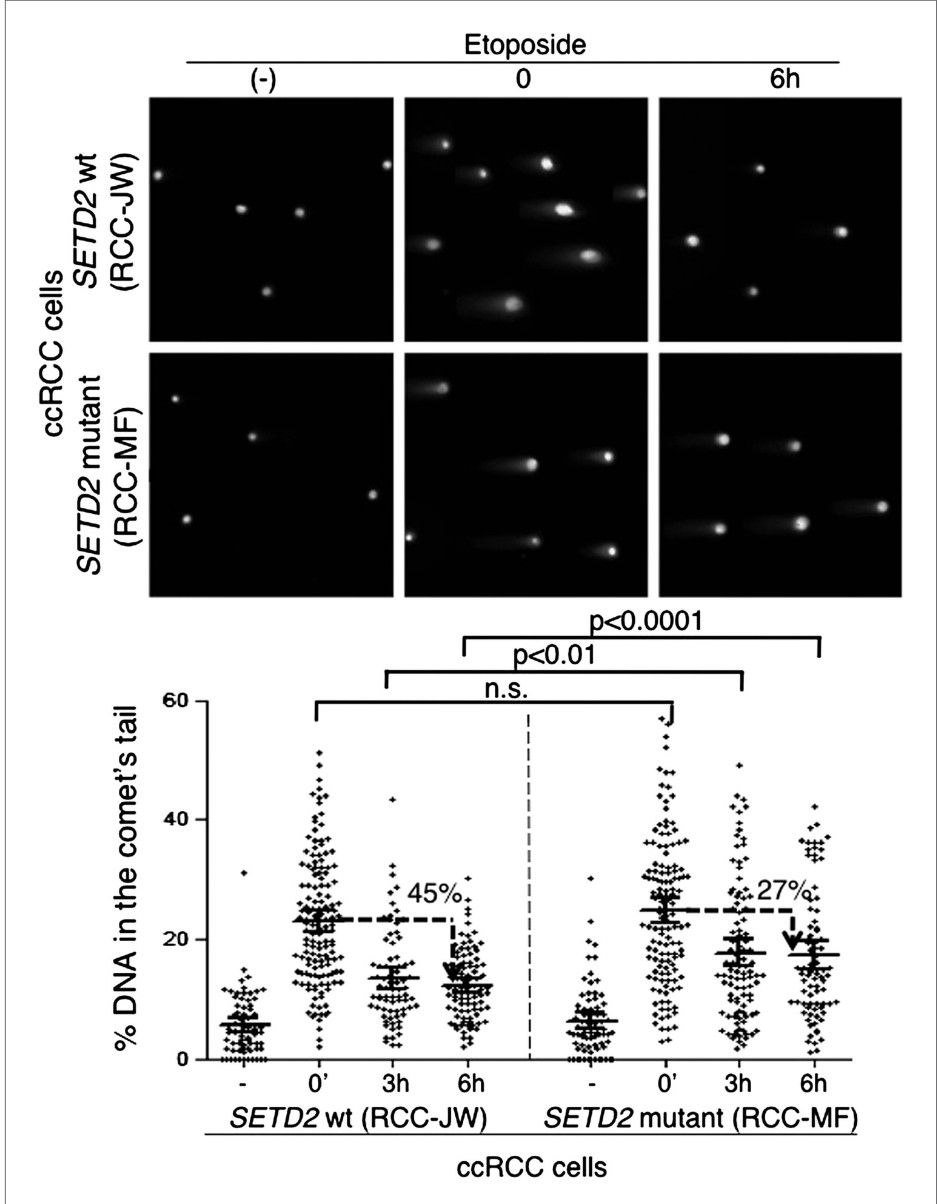

**Figure 7**. Repair of double-strand breaks is compromised in *SETD2* mutant ccRCC cells. The amount of DNA damage immediately after induction of DSBs with etoposide and 3 and 6 hr after treatment was estimated by comet assay on *SETD2* wt and mutant ccRCC cells. Graphs depict the % of DNA in the tail of each individual comet. The horizontal solid lines represent the mean and 95% confidence intervals. At least 70 individual comets were analyzed for each time point.

The following figure supplements are available for figure 7:

**Figure supplement 1**. Repair of double-strand breaks in SETD2 wt and mutant ccRCC cells.

and to initiate signal transduction cascades that activate p53 and inactivate cyclin-dependent kinases causing inhibition of cell cycle progression (*Sancar et al., 2004*). The tumor suppressor p53 is a direct substrate of ATM kinase (*Canman et al., 1998*) and is a master effector of the DNA damage cell cycle checkpoints (*Sperka et al., 2012*). Under normal conditions, p53 protein levels are low owing to MDM2-mediated ubiquitylation and proteasomal degradation. Upon DNA damage, p53-phosphorylation inhibits its interaction with MDM2, resulting in p53 stabilization (*Shieh et al., 1997*; *Banin et al., 1998*). We therefore evaluated both p53 phosphorylation and its total protein levels in response to

DNA damage. The two *SETD2* wt ccRCC cell lines (RCC-JW and Caki-2) responded to DSBs with p53 phosphorylation and increased total p53 protein levels 2 hr after initiating the DDR (*Figure 8A*). In contrast, despite the persistent DNA damage caused by the impaired DDR, *SETD2* mutant ccRCC cells (RCC-MF and RCC-FG2) evaded the cell cycle checkpoint as revealed by the low levels of p53 phosphorylation and unnoticeable change of its total protein levels (*Figure 8A*). Activation of the p53 checkpoint leads to increased gene expression levels of several downstream targets including *CDKN1A*, also known as *p21* (*Sperka et al., 2012*). Indeed, p21 protein levels rose significantly 2 hr after DNA damage in *SETD2* wt cells, but not in their mutant counterparts (*Figure 8A*). To exclude the contribution of SETD2-independent mechanisms due to the different genetic backgrounds of the *SETD2* wt and mutant cells, we measured p53 checkpoint activation in *SETD2* RNAi cells upon treatment with etoposide, NCS or phleomycin (*Figure 8B–D*). *SETD2* RNAi-depleted ccRCC cells phenocopied the severed p53 cell cycle checkpoint observed in *SETD2* mutant ccRCC cells. In response to each of the three DNA-damaging agents, control cells increased the total levels of p53 and p21, whereas SETD2-depleted cells failed to activate p53 (*Figure 8B–D*).

To further examine whether SETD2 is necessary for DNA damage-induced cell-cycle checkpoints, we performed flow cytometry-based cell-cycle profiling analyses in control and *SETD2* RNAi depleted U2OS cells. Cells were treated with phleomycin to activate G1/S and G2/M DNA damage checkpoints. After phleomycin treatment, 49% (8 hr) and 46% (12 hr) of control cells were arrested at G1 phase and 34% (8 hr) and 38% (12 hr) stayed in G2 phase (*Figure 9A*). In contrast, only 42% (8 hr) and 34% (12 hr)

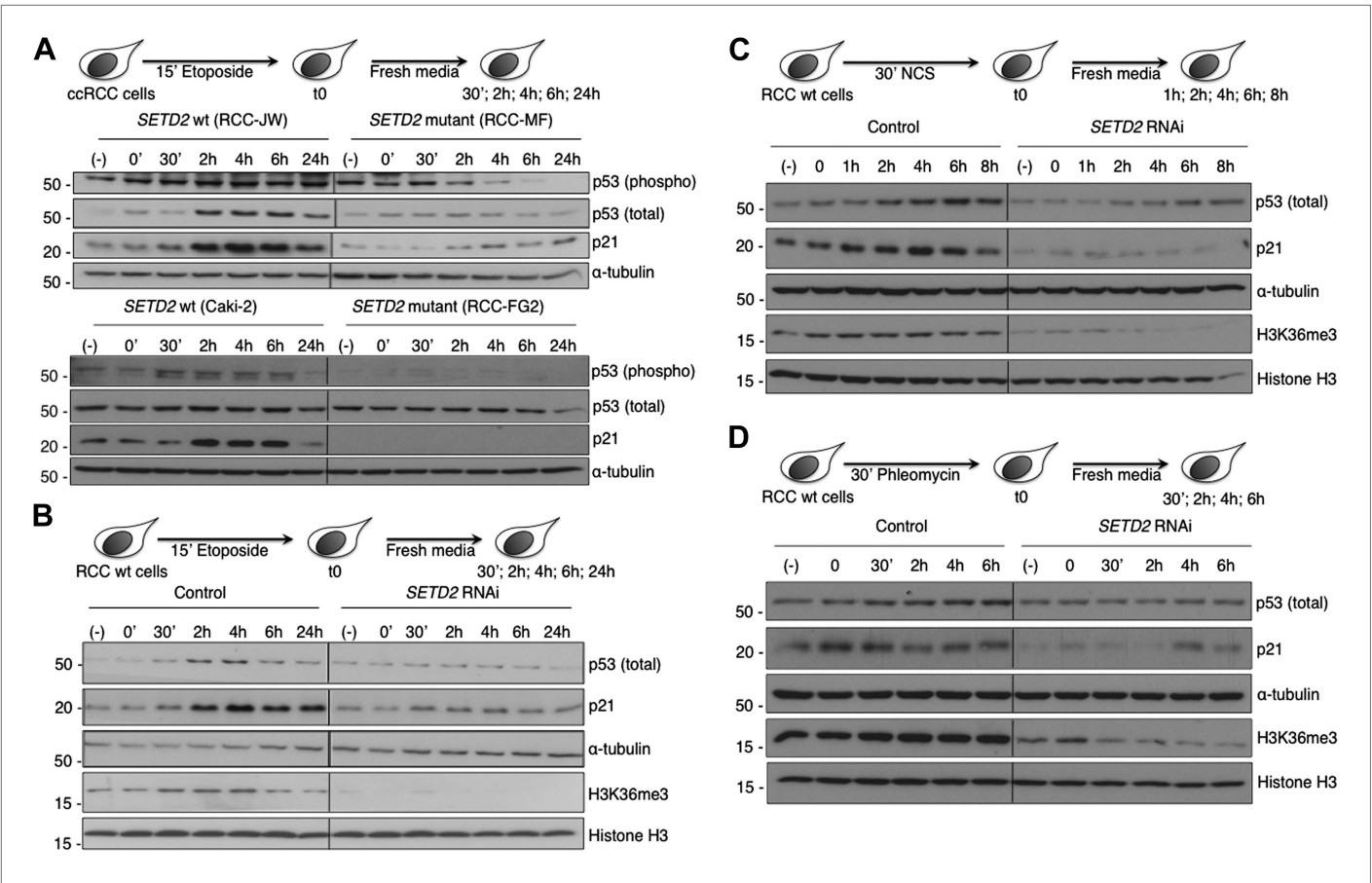

**Figure 8**. SETD2 is required for p53 activation following DNA damage. (**A**) *SETD2* wt and mutant ccRCC cells were challenged with etoposide during 15 min and chased in fresh media during the indicated time points. Western blot analysis was performed with antibodies against the proteins indicated on the right. *SETD2* wt ccRCC cells were depleted for SETD2 by RNAi, treated with etoposide (**B**), NCS (**C**) or phleomycin (**D**) during the indicated periods of time and chased in fresh media. Western blot analysis was performed with antibodies against the proteins indicated on the right. Molecular weight markers (KDa) are shown on the left. Data are from one representative experiments of a total of three independent experiments performed with similar results.

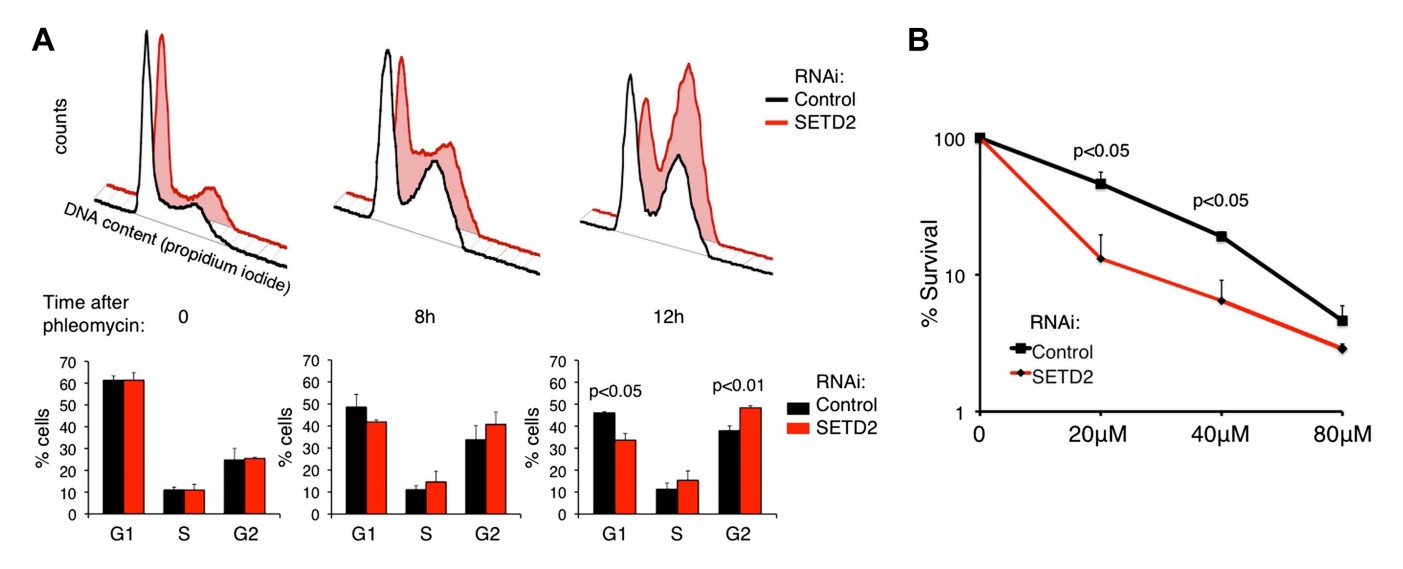

**Figure 9**. SETD2 regulates the p53-dependent cell-cycle checkpoint and cell survival following DNA damage. (**A**) Control and *SETD2* RNAi-depleted cells were treated with 40 μM of phleomycin to induce cell-cycle checkpoints. Cells were harvested at 0, 8, or 12 hr after treatment, fixed and stained with propidium iodide, and analyzed by flow cytometry. Graphs show means and standard deviations of the percentage of cells at each phase of the cell cycle. Data are from four individual experiments. (**B**) Control and *SETD2* RNAi-depleted cells were treated with phleomycin at indicated concentrations and clonogenic survival was measured 10 days after treatment. Means and standard deviations from 6 individual cell cultures treated with each drug concentration are shown.

of cells were arrested in G1 phase, whereas 41% (8 hr) and 48% (12 hr) were at G2 phase upon SETD2 depletion. These results suggest that SETD2 depletion leads to a defective G1/S cell-cycle arrest, which is in agreement with the deficient activation of p53.

To investigate whether SETD2 affects the cell survival rate in response to DNA damage, we depleted U2OS cells for *SETD2* by RNAi and tested their capacity for clonogenic survival after phleomycin treatment (*Figure 9B*). 10 days after DNA damage, the colony-forming capacity of control cells was significantly higher than SETD2-depleted cells. This result suggests that SETD2 is a determinant of cell survival after DNA damage.

Combined, these data point to a novel role of SETD2 that can explain the puzzling observation that ccRCC often evade the p53 cell-cycle checkpoint despite a very low frequency of *TP53* mutations (*Gurova et al., 2004*; *Dalgliesh et al., 2010*; *Sato et al., 2013*).

## Discussion

In this study, by focusing on the pathways that repair DSBs upon RNAi depletion of *SETD2*, we reveal that this histone methyltransferase is necessary for the DDR. We show that SETD2 is required for the activation of ATM following DNA damage. In agreement with the impaired ATM activation, spreading of γH2AX and 53BP1 phosphorylation was greatly compromised in SETD2-depleted cells treated with NCS or etoposide, but only marginally affected in DSBs induced by phleomycin. These findings suggest that the DNA damage signaling may follow multiple pathways with different requirements for H3K36me3 and can result in phosphorylation of H2AX and 53BP1 independently of ATM activation. In fact, ionizing radiation-induced H2AX phosphorylation and recruitment of 53BP1 to DSBs can be carried out by ATM and DNA-dependent protein kinase (DNA-PK) in a redundant manner (*Stiff et al., 2004*). Moreover, ataxia telangiectasia and Rad3-related protein (ATR) can also phosphorylate many of the ATM substrates, including H2AX (*Ward and Chen, 2001*).

53BP1 nucleation at DNA-damaged chromatin plays an important role in favoring DNA repair by NHEJ (*Bouwman et al., 2010*; *Bunting et al., 2010*; *Chapman et al., 2012*). In agreement, we found that induction of DSBs resulted in a significant impairment of the 53BP1 recruitment and increased DNA 5′ end resection in SETD2-depleted cells. However, the increased resection did not reflect increased efficiency of HR repair. Instead, persistent DNA 5′ end resection is likely due to the roadblock in the

HR pathway that is created by the failure of RAD51 binding to the single-stranded DNA. Indeed, this defect in presynaptic filament formation on the resected DNA resulted in a 50% reduction in the efficiency of gene conversion of the DR-GFP HR reporter. Exonuclease-mediated resection of DNA ends is known to promote phosphorylation of RPA and to orchestrate an ATM-to-ATR switch at DNA breaks (*Shiotani and Zou, 2009*). This switch in the main DNA damage sensor provides a reasonable explanation for the RPA phosphorylation and the proficient RPA recruitment to DSBs despite the impaired ATM activation in SETD2-depleted cells. Alternatively, ATM-independent RPA phosphorylation can also be achieved through the activity of DNA-PK (*Shao et al., 1999*; *Anantha et al., 2007*).

Lens-epithelium-derived growth factor p75 splice variant (LEDGF) is an H3K36me3-binding protein that promotes the HR repair of DSBs (*Daugaard et al., 2012*). Depletion of LEDGF impairs the recruitment of CtIP and the subsequent DNA 5′ end resection of DSBs. While defects in LEDGF recruitment to DSBs may account for the impaired HR phenotype that we report in SETD2 deficient cells, our data supports an extended LEDGF-independent role of SETD2 in the repair of DSBs by HR. Our findings that DNA 5′ end resection and RPA foci formation are not severely compromised in SETD2-depleted cells disclose a specific presynaptic role for SETD2 that is necessary for the recruitment of RAD51 to single-stranded DNA. One possibility is that H3K36me3 defines a local chromatin state that favors the recruitment of RAD51 to DSBs and successful HR repair. This could be achieved by the FACT histone chaperone upon recruitment to H3K36me3-marked nucleosomes (*Carvalho et al., 2013*). In agreement with this view, depletion of the FACT subunit SPT16 results in a major impairment of HR (*Kari et al., 2011*; *Oliveira et al., 2014*). FACT is required to recruit the ubiquitin ligase RNF20, which in turn facilitates RAD51 accumulation at DSBs (*Oliveira et al., 2014*). The view that changes in chromatin dynamics may affect RAD51 function is not completely unexpected as illustrated by the role of the histone acetyltransferase 1 (HAT1) in RAD51 during HR (*Yang et al., 2013*). HAT1 promotes acetylation of histone H4, as well as incorporation of histone variant H3.3 at DSBs and facilitates RAD51 recruitment and efficient HR repair. Nevertheless, the major impact of SETD2 on RNA polymerase II transcription and pre-mRNA processing that others and we have previously described (reviewed in *de Almeida and Carmo-Fonseca, 2014*) discloses the alternative possibility that the impaired formation of RAD51 filaments and HR may result from disrupted gene expression of key DNA repair factors. Hence, additional studies are needed to fully elucidate the mechanistic basis of the SETD2 impact on the DDR.

Unlike all histone modifications found to promote the DDR, previous studies (*Pei et al., 2011*) and our data show no evidence of increased H3K36me3 levels at the DNA damage sites or of a DDR-dependent recruitment of SETD2 to DSBs. These findings disclose an unprecedented role for a histone-modifying enzyme in priming intact chromatin for the response to DNA damage. A model that emphasizes the regulation of the H3K36me3 accessibility as a major determinant of the DDR can be envisaged. According to this view, the conformational changes that occur after a DSB may cause exposure of H3K36me3 marks previously written by SETD2 in intact chromatin, for instance during transcription by RNA polymerase II or replication. In fact, a careful inspection of the nucleosome structure reveals that lysine 36 on the N-terminal tail of histone H3 does not protrude from the nucleosome core. Instead, this lysine is hidden by the DNA double strand that wraps the histone octameric core (*Luger et al., 1997*). This sheltered location was first described three decades ago, when Crane–Robinson et al. investigated the accessibility of the histone-terminal domains to proteases. Their experiments revealed that while the first 26 N-terminal residues of histone H3 were greatly sensitive to trypsin digestion, H3K36 was resistant to the enzyme (*Bohm et al., 1981*). According to this 'break and access model' DNA damage exposes H3K36me3, which then contributes together with a set of well-defined histone modifications to the orchestration of the chromatin landscape that drives the initial recruitment and activation of DDR factors, such as ATM.

The role of H3K36me3 in promoting repair of DSBs by HR may be particularly relevant at coding regions, where the introduction of genomic aberrations is likely to have deleterious consequences. Therefore, in these regions the choice between error-prone NHEJ and the more reliable HR pathway is of utmost importance. In fact, most H3K36me3 is found at actively transcribed chromatin where it signals the recruitment of the histone chaperone FACT to reassemble nucleosomes in the wake of RNA polymerase II elongation (*Carvalho et al., 2013*). Should a DSB amputate a coding region, exposure of H3K36me3 previously deposited by SETD2 traveling with the transcriptional machinery may promote DNA repair by HR and would guarantee the integrity of the genetic information.

From all cancers on which *SETD2* mutations have been reported, ccRCC shows the highest mutation rate (*Dalgliesh et al., 2010*; *Varela et al., 2011*). Our present findings reveal that *SETD2* mutant ccRCC cells have impaired DDR signaling which leads to inefficient repair of DNA damage. However, the

p53-dependent cell cycle checkpoint is not activated in SETD2-deficient cells despite the persistence of DNA damage. Depletion of SETD2 abrogates p53 activation, and leads to eviction of the G1/S checkpoint upon DNA damage. In addition to the defects in the DNA damage signaling that we found in SETD2-depleted cells, which could extend to improper p53 activation, it is also possible that SETD2 impinges directly on p53 activity and activation of its downstream targets through direct protein–protein interactions (*Xie et al., 2008*). Furthermore, in the absence of SETD2, the capacity for clonogenic survival following DNA damage is significantly reduced. These findings integrate SETD2 into the p53 regulatory network and provide a reasonable explanation for the intriguing observation that inactivation of this master regulator of cell cycle checkpoints in ccRCC is rarely caused by mutations in *TP53* (*Gurova et al., 2004*; *Dalgliesh et al., 2010*; *Sato et al., 2013*).

A plethora of recent studies have revealed that mutation rates and genomic instabilities in cancer are directly related to aberrant histone modifications and chromatin organization, underscoring the importance of the histone code in maintaining genome stability (*Lukas et al., 2011*; *Papamichos-Chronakis and Peterson, 2013*). CcRCC is an example of a cancer, where a high rate of genomic alterations occurs in parallel to inactivating mutations and deletions of histone modifiers, such as *SETD2* (*Cancer Genome Atlas Research, 2013*; *Sato et al., 2013*). The present study provides a functional link between these two events extending our understanding of the impact of altered chromatin states on the mechanisms that maintain genome integrity. Although it is now evident that alterations in histone modifications are a hallmark of most cancers, it is still unclear whether these changes are drivers or passengers of tumorigenesis (*Waldmann and Schneider, 2013*). Our findings endorse the classification of SETD2 as a tumor suppressor and place inactivating mutations in this histone modifier at the peak of a harmful cascade of events that may ultimately drive malignant cell transformation and tumor development.

## Materials and methods

### Cell culture and drug treatments

Human osteosarcoma (U2OS) cells (ATCC) and the ccRCC cells RCC-JW, Caki-2, RCC-MF and RCC-FG2 (*Duns et al., 2010*) (Cell Line Services Eppelheim, Germany) were grown as monolayers in Dulbecco's modified Eagle medium–DMEM (Invitrogen, Carlsbad, CA), supplemented with 10% (vol/vol) FBS, 1% (vol/vol) nonessential amino acids, 1% (vol/vol) L-glutamine, and 100U/ml penicillin-streptomycin, and maintained at 37°C in a humidified atmosphere containing 5% $CO_2$. To induce DNA damage, cells were treated with 50 µM etoposide (Sigma, St Louis, MI) for 15 min, 250 ng/ml neocarzinostatin (NCS) (Sigma) for 30 min or 40 µM phleomycin (Sigma) for 30 min, washed and harvested immediately after the treatment and at the indicated time-points.

### RNA interference

*SETD2* RNAi was achieved using three different synthetic siRNA duplexes targeting *SETD2* (Eurogentec, Seraing, Belgium) (*Carvalho et al., 2013*). In each experiment, at least two different siRNAs were used resulting in similar knockdown efficiencies. siRNAs targeting the firefly luciferase (GL2) were used as controls (Eurogentec) (*Elbashir et al., 2001*). Cells were reverse transfected with 10 µM siRNAs using OptiMEM (Invitrogen) and Lipofectamine RNAiMAX (Invitrogen), according to the manufacturer's instructions. 24 hr after the first transfection, cells were re-transfected with the same siRNA duplexes and transfection reagents and harvested on the following day.

### SETD2-GFP plasmid construction

The *SETD2* cDNA was cloned into the pEGFP-C1 vector (Clontech, Mountain View, CA) in a ligation reaction containing two fragments. A fragment containing 5257 bp from the 3′ end of *SETD2* was obtained by digestion of pCR-XL-TOPO-SETD2 (clone 40125713; Thermo Scientific, Waltham, MA) with *Spe*I and *Pst*I. A fragment containing the 5′ end of *SETD2* was obtained by PCR amplification of pCR-XL-TOPO-SETD2 using primers containing *Bgl*II and *Spe*I restriction sites. The PCR product was digested with *Bgl*II and *Spe*I restriction enzymes and purified. Both fragments were co-inserted in *Bgl*II/*Pst*I digested pEGFP-C1 vector. This construct was confirmed by DNA sequencing.

### Immunofluorescence

U2OS cells grown on coverslips were fixed with 3.7% paraformaldehyde for 10 min at room temperature. The cells were then permeabilized with 0.5% Triton X-100/PBS for 10 min. Incubation with primary

antibodies against γH2AX (phosphoSer139; 05–636, Millipore, Billerica, MA), total RPA32 (ab2175; Abcam, Cambridge, UK), BRCA1 (sc-6954; Santa Cruz Biotechnology, Dallas, TX), RAD51 (ab213; Abcam), or cyclin A (sc-751; Santa Cruz Biotechnology) was followed by incubation with fluorochrome-conjugated (Alexa Fluor 488 and 594) antibodies (Jackson Immunoresearch, West Grove, PA). All the washing steps were done with PBS containing 0.05% (vol/vol) Tween 20. The samples were mounted in Vectashield (H-1000; Vector Laboratories, Burlingame, CA) with 4'-6-diamidino-2-phenylindole (Dapi) (09542; Sigma) to stain the DNA. A Zeiss LSM 710 (Carl Zeiss, Oberkochen, Germany) confocal microscope was used to visualize the cells with a 63x/1.4 oil immersion or a 40x objective.

## Western blot

Whole cell protein extracts were prepared by cell lysis with SDS-PAGE buffer (80 mM Tris-HCL pH 6.8, 16% glicerol, 4.5% SDS, 450 mM DTT, 0.01% bromophenol blue) with 200U/ml benzonase (Sigma) and 50 μM $MgCl_2$ and boiling for 5 min. Equal amounts of protein extracts were resolved by SDS-polyacrylamide gel electrophoresis (SDS-PAGE) and transferred to a nitrocellulose membrane. Immunoblotting was performed with antibodies against the following proteins: H3K36me3 (ab9050; Abcam), histone H3 (ab1791; Abcam), phospho-53BP1 (phosphoSer1778; No.2675, Cell Signaling, Danvers, MA), phospho-ATM (phosphoS1981; No.200-301-400s, Rockland, Gilbertsville, PA), total ATM (PC116; Millipore), total RPA32 (ab2175; Abcam), phospho-RPA32 (phosphoSer4/Ser8; A300-245A, Bethyl, Montgomery, TX), α-Tubulin (T5168; Sigma); γH2AX (phosphoSer139; 05-636, Millipore); Histone H2B (ab1790; Abcam); GFP (11814460001; Roche, Basel, Switzerland); total p53 (sc-263; Santa Cruz); phospho-p53 (phosphoSer15; ab38497, Abcam); p21 (sc-397; Santa Cruz).

## Laser-induced DNA damage

Laser-induced DSBs were generated using a confocal microscope (LSM 510 Meta, Carl Zeiss) equipped with a 37°C heating chamber (Pecon, Erbach, Germany) and a 405 nm diode laser focused through a 63x/1.4 oil immersion objective. We performed one iteration at a laser output of 100%. To follow the dynamics of GFP-tagged 53BP1 (53BP1-GFP, a kind gift from Dr Jiri Lukas, Danish Cancer Society, Copenhagen, Denmark) or SETD2-GFP during the DDR, live-cell images were recorded during 15 min after laser-irradiation. Control or SETD2-depleted U2OS cells were transfected with expression plasmids for 53BP1-GFP or SETD2-GFP using Lipofectamine 2000 (Invitrogen) according to the manufacturers's protocol.

## Nuclear co-immunoprecipitation

Briefly, nuclear extracts and further MNase digestion was performed as described earlier (*Carvalho et al., 2013*) with minor changes. Cells were lysed in ice-cold NP-40 buffer (10 mM Tris–HCl [pH 7.4], 10 mM NaCl, 3 mM $MgCl_2$, 0.5% Nonidet P-40, 0.15 mM spermine, and 0.5 mM spermidine) and isolated nuclei were washed and resuspended in MNase digestion buffer (10 mM Tris–HCl [pH 7.4], 15 mM NaCl, 60 mM KCl, 0.15 mM spermine, and 0.5 mM spermidine) and sonicated with a single pulse of 15 s at 50% intensity using a Soniprep 150 (Sanyo, Moriguchi, Japan). Nuclei were digested enzymatically with 60U MNase (Thermo Scientific) at 4°C for 60 min. MNase stop buffer (100 mM EDTA and 10 mM EGTA [pH 7.5]) was used to stop the reaction and samples were pre-cleared using Protein G Dynabeads (Life Technologies, Carlsbad, CA) at 4°C for 45 min. All buffers were supplemented with protease and phosphatase inhibitors (Roche). Samples were incubated with 3 μg of either anti-53BP1 (ab87097; Abcam) or anti-H3K36me3 (ab9050; Abcam) antibodies overnight at 4°C. The protein complexes were pulled down using Protein G Dynabeads, washed six times in IP buffer (200 mM NaCl, 16.7 mM Tris pH 8.1, 1.1% TritonX-100, 1.2 mM EDTA, 0.01% SDS) and resolved by SDS-PAGE before immunoblotting with antibodies against: γH2AX (phosphoSer139; 05-636, Milipore) and anti-H3K36me3 (ab9050; Abcam). 1/10th of the total cell lysate was used as input samples.

## Homologous recombination reporter assay

The U2OS DR-GFP cells were a kind gift of Dr Maria Jasin from the Memorial Sloan-Kettering, USA. To assess HR efficiency, U2OS DR-GFP cells were seeded in a six-well plate and reverse transfected with the *I-SceI* expression plasmid (pCBA-SceI; 26477; Addgene, Cambridge, MA) using Lipofectamine 2000 (Invitrogen), according to the manufacturers's protocol. The cells were maintained for 24 hr at 37°C in a humidified atmosphere containing 5% $CO_2$. After 24 hr, siRNAs targeting either firefly luciferase (GL2) or *SETD2* were transfected using Lipofectamine RNAiMAX (Invitrogen). 48 hr after RNAi (and 72 hr after *I-SceI* transfection) cells were harvested and subjected to flow cytometry analysis (FACS Calibur;

BD Biosciences, San Jose, CA) with a minimum of 20,000 events acquired. The enzyme cutting efficiency (35% on control cells and 33% on SETD2-depleted cells) was determined by real-time quantitative PCR (RT-qPCR) with primers flanking the *I-Sce*I cleavage site on the *GFP* reporter gene. The amount of DNA was estimated as follows: $2^{\wedge}(\text{Ct reference–Ct sample})$, where Ct reference and Ct sample are mean threshold cycles of RT-qPCR done in duplicate from *GAPDH* (reference) and the DNA flanking the *I-Sce*I site (sample). Non-transfected cells were used as a reference to calculate the percentage of cutting upon *I-Sce*I transfection. Primers are detailed in *Table 1*. The percentage of GFP+ cells was estimated using FlowJo (TreeStar, Ashland, OR).

## Cell cycle analysis and survival assays

The nuclear DNA content was estimated by flow cytometry analysis of cells stained with propidium iodide (PI). Briefly, cultured cells were collected by trypsinization and fixed by drop wise addition of ice-cold 50% ethanol with gentle vortexing. After placing the samples 24 hr on ice, RNase was added to a final concentration of 50 µg/ml and incubated for 30 min at 37°C. Following addition of PI (100 µg/ml), 80,000 events were acquired on a FACS Calibur (BD Biosciences) and the DNA content analyzed with FlowJo (TreeStar). For analysis of cell cycle checkpoints, control or SETD2-depleted U2OS cells were treated with 40 µM phleomycin for 30 min, washed and cultured in fresh medium for 8 and 12 hr before flow cytometry analysis of the cell cycle progression. For analysis of clonogenic survival, cells were reverse-transfected with SETD2 and control (GL2) siRNAs, seeded in 10-cm dishes (1000 cells/dish) and 2 days later treated for 30 min with phleomycin at various concentrations. Cells were then cultured for 10 days and individual colonies counted upon staining with crystal violet.

## Direct measurement of DNA 5′ end resection

The extent of DNA 5′ end resection was analyzed by RT-qPCR analysis of the DNA surrounding the *I-Sce*I site of the *GFP* gene integrated in the U2OS cells (U2OS DR-GFP reporter) transfected with the *I-Sce*I expression plasmid. Primers flanking the *I-Sce*I site and four regions located approximately 250 bp, 300 bp, 350 bp and 2200 bp away from this site were used. The housekeeping gene *GAPDH* was used as a genomic DNA control. RT-qPCR was performed in the ViiA Real Time PCR (Applied Biosystems, Foster City, CA) using Power SYBR Green PCR Master Mix (Applied Biosystems). The percentage of digestion was estimated as follows: $2^{\wedge}(\text{Ct reference–Ct sample})$, where Ct reference and Ct sample are mean threshold cycles of RT-qPCR done in duplicate on DNA samples from *GAPDH* (reference) and the DNA regions surrounding the *I-Sce*I site (sample). The amount of DNA was corrected for the *I-Sce*I digestion efficiency by normalizing the data against the amount of DNA flanking the restriction site. All primer sequences are presented in *Table 1*.

**Table 1.** Sequence of primers used in this study

| Gene Name | Primer designation | Primer sequence |
| --- | --- | --- |
| *I-Sce*I *Flank* | Flank_Fw | GAG CAG GAA CCT GAG GAG |
| | Flank_Rv | GCC GTA GGT ATT ACC CTG |
| *I-Sce*I *250bp* | 250bp_Fw | CAT GCC CGA AGG CTA CGT |
| | 250bp_Rv | CGG CGC GGG TCT TGT A |
| *I-Sce*I *300bp* | 300bp_Fw | CTT CTT CAA GGA CGA CGG C |
| | 300bp_Rv | GTA GTT GTA CTC CAG CTT GTG |
| *I-Sce*I *350bp* | 350bp_Fw | CTA CAG CTC CTG GGC AAC |
| | 350bp_Rv | CTT CGG CAC CTT TCT CTT C |
| *I-Sce*I *2200bp* | 2200bp_Fw | CGC GAC GTC TGT CGA GAA G |
| | 2200bp_Rv | GCC GAT GCA AAG TGC CGA TA |
| GAPDH | GAPDH_Fw | GAA GGT GGA GGT CGG AGT C |
| | GAPDH_Rv | GAA GAT GGT GAT GGG ATT TC |

## Comet assays

DNA double-strand breaks in *SETD2* wt or mutant ccRCC cells were detected by neutral comet assay using the CometAssay kit (Trevigen, Gaithersburg, MD) according to the manufacturer's instructions. A Zeiss Axiovert 200M (Carl Zeiss) microscope equipped with a 20x objective was used to visualize cells. The amount of DNA damage was estimated by measuring the percentage of fluorescence in the comet's tail using the CometScore analysis software.

## Statistical analysis

Where indicated, data were subjected to a two-tailed Student's t-test to resolve statistical significance.

## Acknowledgements

We thank our colleagues Dr Maria Carmo-Fonseca and Dr Miguel Godinho-Ferreira for critical comments and suggestions. We thank Ana Rita Grosso, Ana Luísa Caetano, Mafalda Matos, Sérgio Marinho, and Pedro Pereira for technical assistance. We are grateful to Dr Maria Jasin and Dr Jiri Lukas for kindly providing the U2OS DR-GFP cells and 53BP1-GFP expression plasmid, respectively. We also thank the Bioimaging and Flow Cytometry facilities of the IMM for technical assistance. This work was supported by grants from Fundação para a Ciência e Tecnologia (FCT), Portugal (PTDC-BIA-BCM-111451-2009 and PTDC/BIM-ONC/0384-2012 to SFdA). ACV is a Lisbon BioMed PhD fellow funded by FCT (SFRH/BD/52232/2013). SCS is supported by RNPnet, a Marie Curie Initial Training Network (PITN-GA-2011-289007). JF receives support from a Gulbenkian Foundation grant (96526/2009). SFdeA is the recipient of a FCT Investigator award.

## Additional information

### Funding

| Funder | Grant reference number | Author |
| --- | --- | --- |
| Fundação para a Ciência e Tecnologia, Portugal | PTDC-BIA-BCM-111451-2009 | Sérgio F de Almeida |
| Fundação para a Ciência e Tecnologia, Portugal | PTDC/BIM-ONC/0384-2012 | Sérgio F de Almeida |
| Marie Curie Initial Training Network (RNPnet) | PITN-GA-2011-289007 | Sreerama C Sridhara |
| Fundação para a Ciência e Tecnologia | SFRH/BD/52232/2013 | Alexandra C Vítor |
| Fundação Calouste Gulbenkian | 96526/2009 | João Ferreira |

The funders had no role in study design, data collection and interpretation, or the decision to submit the work for publication.

### Author contributions

SC, ACV, SFA, Conception and design, Acquisition of data, Analysis and interpretation of data, Drafting or revising the article; SCS, FBM, ACR, Acquisition of data, Analysis and interpretation of data; JMPD, Analysis and interpretation of data, Contributed unpublished essential data or reagents; JF, Acquisition of data, Analysis and interpretation of data, Drafting or revising the article

### Author ORCIDs

Sérgio F de Almeida, http://orcid.org/0000-0002-7774-1355

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
