## [Decision Letter]

Thank you for sending your work entitled “SETD2 is required for DNA double-strand break repair and activation of the p53-mediated checkpoint” for consideration at *eLife*. Your article has been favorably evaluated by a Senior editor and 3 reviewers, one of whom is a member of our Board of Reviewing Editors.

The Reviewing editor and the other reviewers discussed their comments before we reached this decision, and the Reviewing editor has assembled the following comments to help you prepare a revised submission.

In this interesting paper, Carvalho et al report a novel role for the lysine methyl-transferase SETD2 in the DNA damage response (DDR). They demonstrate that SETD2 depletion impairs several key signaling events downstream of DNA double-stranded breaks, including phosphorylation of ATM, H2A.X and 53BP1, recruitment of 53BP1 to sites of DNA damage, and formation of Rad51 foci, but not RPA recruitment. Depletion of SETD2 also produces a defect in DNA repair via homologous recombination (HR) and a defect in p53 activation after DNA damage. These findings are also reproduced in clear cell renal carcinoma cell lines carrying SETD2 mutations. The reviewers found these findings to be of high impact and significant, because loss-of-function mutations in SETD2 have been recently documented in various malignancies, suggesting that SETD2 is a bona fide tumor suppressor. The findings in this report may explain, at least in part, how SETD2 suppresses tumor growth.

However, the reviewers discussed how the mechanistic aspects of the paper could have been more developed. In particular, the reviewers noted the difficulty in dissecting direct effects (i.e., SETD2 acting at the sites of DNA damage via H3K36 methylation) versus indirect SETD2 action (i.e., via transcriptional effects). In the end, the reviewers agreed that the paper would merit publication in a high impact journal such as *eLife* by including additional experiments and controls that would significantly strengthen the main conclusions, even if further mechanistic insight was not provided.

Therefore, the major concerns that will have to be addressed for this manuscript to be considered for publication are:

1) A key conclusion in the manuscript, stated in the title, is that SETD2 is involved in the 'p53 activation checkpoint'. However, this is based merely on p53 and p21 Western blots. Although p53 and p21 levels are lower in SETD2 deficient cells, no work is performed on the checkpoint itself so the title of the figure is misleading and overstated. A direct measurement of the G1 checkpoint should be performed with and without SETD2 to strengthen this statement. Cells can be damaged with IR (or phleo, bleo if not available) followed by a time course using FACS analysis to measure DNA content and G1 arrested cells. siControl and siSETD2 cells can be compared to see if indeed depletion of SETD2 affects the p53-dependent checkpoint at the level of the cell cycle as reported by the authors. In addition, clonogenic DNA damage sensitivity assays with these agents would help show that these defects observed in SETD2 defective cells actually have some biological consequence. As a suggestion, Etoposide should not be used as this response is dependent on TOP2, which might be altered in SETD2 depleted cells. Although this could be interesting, the results would be indirect and these experiments are suggested to look at direct effects of SETD2 upon DNA damage on the p53 checkpoint (see point 2).

2) Many of the key experiments inducing DNA double-stranded breaks were done with etoposide, a topoisomerase II inhibitor (such as phospho-H2AX formation). Since TOP2 is involved in transcription, it is important to repeat these experiments with other agents that cause double-stranded breaks by other mechanisms, such as neocarcinostatin (NCS), and ionizing radiation (IR, or a radiomimetic drug such as phleomycin or bleomycin). To what the degree the defects in DNA damage-induced signaling observed in SETD2-depleted cells can be reproduced with these other agents? More specifically, the reviewers would like to know if other signaling events, such as ATM, H2A.X, 53BP1 and p53 activation are also impaired in SETD2-depleted cells after treatment with these other agents.

3) Figure 4 does not provide enough information for the results to be judged. Information about what time point this data represents and what the cutting efficiency is should be provided. Additionally, in Figure 4, all cells show similar amounts of RPA after NCS. G1 cells should not show RPA similarly to S/G2 cells since NHEJ is the more prominent DNA repair pathway. Are all of these cells in S/G2? No cell cycle analysis beyond a FACS profile is provided and since resection, DNA repair and DNA damage signaling are regulated throughout the cell cycle, a more careful analysis is warranted. This can be accomplished by using Cyclin A staining in the IF analysis to indicate S/G2 cells and G1 cells that will lack Cyclin A staining. No information is provided for why RAD51, but not RPA, are defective in SETD2 deficient cells. Could the authors discuss a possible mechanism? LEDGF has been shown to bind transcription marks including H3K36me3 and be involved in DNA repair (Daugaard et al. NSMB, 2012) but how SETD2 relates to this work is not discussed. Can you detect defects in CTIP, BRCA1 and/or LEDGF recruitment to damage sites in the absence of SETD2? This data would allow a deeper understanding of what aspect of HR are regulated by SETD2.

4) The immunofluorescence-based DNA damage assays such in Figure 1 (γH2AX) and Figure 3 (RPA) should be performed in a quantitative manner, similar to that of Figure 3.

5) The experiments attempted to address the if K36me3 is present at the nucleosomes adjacent to DSBs (Figure 2) are problematic in several aspects:

a) To prepare mono-nucleosomes, the common nuclease of choice is MNase. However, the authors used DNase I, which efficiently cleave nucleosomal DNA at minor grooves, and Benzonase, which is even less proven enzyme to specifically digest linker DNA. More importantly, the nuclei DNA digestion pattern indicated a potential over-digestion (DNA fragment is smaller than 147bp), and there is no evidence that nucleosomes remain intact. This experiment should be repeated with MNase.

b) H2A, H2B and H4 western should be included in nucleosome pull-down when H3 antibodies are used.

c) In control cells (Left), when comparing IP to the input, K36me3 IP seems to put down K4me2 more efficiently than K36me3. This result would suggest that K4me2 co-occupies almost completely with K36me3, which is NOT consistent with numerous mass-spec studies and genome-wide analysis. The authors should elaborate on this result.

6) To rule out indirect transcriptional effect caused by Setd2 knockdown, the authors settled to merely analyze the published RNA-seq data in Set2 depleted cells. However, this is not a correct analysis for this purpose. Gene expression comparison should have been done between WT and Setd2-KD cells under DNA damage induced conditions. This analysis should be done in WT and Setd2-KD cells under DNA damage induced conditions or removed from the manuscript entirely. If the authors decide to remove this section, they should acknowledge in the text the possibility of indirect SETD2 action via undefined transcriptional effects.

7) The “Break & access model” proposed by authors in the Discussion section makes no sense to the reviewer for the following reasons:

a) Recombinant Setd2 homologs in human, *Drosophila,* and yeast all show stronger activity toward nucleosomal substrates than histones; b) The works from the Briggs' lab (Du et al 2008 G&D and 2010 JBC) clearly showed that H2A surface is required for Set2-mediated K36me3; c) Based on the structure, the removal of H2A/H2B bears no influence on “disclose H3K36 substrates for Setd2-dependent trimethylation”. Please modify the model to more accurately depict the results obtained and the current state of knowledge.

Experiments using clear cell renal carcinoma cell lines involve only one of each wild type SETD2 versus mutant SETD2. Given that these cell lines are not isogenic, it is imperative to reproduce the key findings in Figures 5, 6 and 7 in at least one other pair of wild type and mutant SET2D.

---

## [Author Response]

*1) A key conclusion in the manuscript, stated in the title, is that SETD2 is involved in the 'p53 activation checkpoint'. However, this is based merely on p53 and p21 Western blots. Although p53 and p21 levels are lower in SETD2 deficient cells, no work is performed on the checkpoint itself so the title of the figure is misleading and overstated. A direct measurement of the G1 checkpoint should be performed with and without SETD2 to strengthen this statement. Cells can be damaged with IR (or phleo, bleo if not available) followed by a time course using FACS analysis to measure DNA content and G1 arrested cells. siControl and siSETD2 cells can be compared to see if indeed depletion of SETD2 affects the p53-dependent checkpoint at the level of the cell cycle as reported by the authors. In addition, clonogenic DNA damage sensitivity assays with these agents would help show that these defects observed in SETD2 defective cells actually have some biological consequence. As a suggestion, Etoposide should not be used as this response is dependent on TOP2, which might be altered in SETD2 depleted cells. Although this could be interesting, the results would be indirect and these experiments are suggested to look at direct effects of SETD2 upon DNA damage on the p53 checkpoint (see point 2)*.

The reviewers make a valid point and we agree that the data presented in the original submission were insufficient. Following the reviewers’ request, we now provide direct measurements of the G1/S checkpoint upon DNA damage with phleomycin (Figure 9). These additional data support our conclusion that depletion of SETD2 affects the p53-dependent checkpoint at the level of the cell cycle. In addition, we analyzed the clonogenic survival of SETD2-depleted cells treated with phleomycin as requested by the reviewers (Figure 9). These experiments revealed that this histone methyltransferase increases the cell survival rate in response to DNA damage.

*2) Many of the key experiments inducing DNA double-stranded breaks were done with etoposide, a topoisomerase II inhibitor (such as phospho-H2AX formation). Since TOP2 is involved in transcription, it is important to repeat these experiments with other agents that cause double-stranded breaks by other mechanisms, such as neocarcinostatin (NCS), and ionizing radiation (IR, or a radiomimetic drug such as phleomycin or bleomycin). To what the degree the defects in DNA damage-induced signaling observed in SETD2-depleted cells can be reproduced with these other agents? More specifically, the reviewers would like to know if other signaling events, such as ATM, H2A.X, 53BP1 and p53 activation are also impaired in SETD2-depleted cells after treatment with these other agents*.

Following the reviewers’ suggestion the revised manuscript now includes data obtained upon DNA damage with NCS and phleomycin (Figures 1 and 8). These new data show that:

a) Defects in the DNA damage signaling are observed in SETD2-depleted cells independently of the DNA damaging drug, as revealed by the impaired ATM activation (Figure 1).

Upon induction of double-strand breaks with etoposide or NCS, H2AX and 53BP1 were phosphorylated in a SETD2-dependent manner, contrasting with the predominant SETD2-independent phosphorylation observed after treatment with phleomycin (Figure 1). These differences are discussed in the manuscript. In light of these new findings, we decided to remove the γH2AX immunofluorescence staining originally shown on Figure 1, as this does not seem to represent a general effect of SETD2 on the DDR. Instead, the DNA damage signaling appears to follow multiple pathways with different requirements for H3K36me3 that can lead to phosphorylation of H2AX and 53BP1 independently of ATM activation.

b) Activation of p53 in response to etoposide, NCS or phleomycin was consistently affected by depletion of SETD2 (Figure 8).

These data confirm that DNA damage signaling and p53 activation are compromised in SETD2-depleted cells and suggest that alternative mechanisms may drive SETD2-independent H2AX and 53BP1 phosphorylation in response to phleomycin-induced DSBs.

*3)*
Figure 4
*does not provide enough information for the results to be judged. Information about what time point this data represents and what the cutting efficiency is should be provided*.

We now provide additional information related with the data shown on Figure 4 (Figure 5 in the revised manuscript) in the “results” and “materials and methods” sections. The experiment was performed 72h after transfection of I-SceI. The cutting efficiency of the enzyme was 35% (control cells) and 33% (SETD2-depleted cells).

*Additionally, in*
Figure 4*, all cells show similar amounts of RPA after NCS. G1 cells should not show RPA similarly to S/G2 cells since NHEJ is the more prominent DNA repair pathway. Are all of these cells in S/G2? No cell cycle analysis beyond a FACS profile is provided and since resection, DNA repair and DNA damage signaling are regulated throughout the cell cycle, a more careful analysis is warranted. This can be accomplished by using Cyclin A staining in the IF analysis to indicate S/G2 cells and G1 cells that will lack Cyclin A staining*.

We thank the reviewers for this additional control. We have repeated the RPA immunofluorescence using Cyclin A staining in parallel. These new data is shown on Figure 3 and shows that only Cyclin A-positive cells have RPA foci upon DNA damage. Moreover, we have also repeated the RAD51 immunofluorescence with Cyclin A co-staining (Figure 4).

*No information is provided for why RAD51, but not RPA, are defective in SETD2 deficient cells. Could the authors discuss a possible mechanism? LEDGF has been shown to bind transcription marks including H3K36me3 and be involved in DNA repair (Daugaard et al. NSMB, 2012) but how SETD2 relates to this work is not discussed. Can you detect defects in CTIP, BRCA1 and/or LEDGF recruitment to damage sites in the absence of SETD2? This data would allow a deeper understanding of what aspect of HR are regulated by SETD2*.

We now discuss a possible mechanism that can sustain the defect in RAD51 recruitment to DSBs in SETD2-depleted cells despite the lack of apparent alterations on RPA phosphorylation and foci formation. The proposed mechanism involves the recruitment of the histone chaperone FACT by H3K36me3 that in turn promotes the formation of RAD51 filaments on resected DNA ends (see Discussion). Moreover, we provide an integrated view of our novel findings with the role of LEDGF in HR repair. While LEDGF may contribute to the aberrant DDR phenotype that we observe in SETD2-depleted cells, we reason that a novel, LEDGF-independent role of SETD2 is necessary for RAD51 recruitment and HR repair. Following the suggestion made by the reviewers, we analyzed BRCA1 recruitment to DSBs by immunofluorescence (Figure 5). These new data show that formation of BRCA1 foci in response to NCS is not affected by depletion of SETD2, thus supporting our conclusion that SETD2 plays a presynaptic role that is necessary for the recruitment of RAD51 to DSBs.

*4) The immunofluorescence-based DNA damage assays such in*
Figure 1
*(γH2AX) and*
Figure 3
*(RPA) should be performed in a quantitative manner, similar to that of*
Figure 3.

We now include these data on Figure 3 and on the new BRCA1 immunofluorescence (Figure 4). Figure 1 was removed from the revised manuscript (see response to point #2)*.*

*5) The experiments attempted to address the if K36me3 is present at the nucleosomes adjacent to DSBs (*Figure 2*) are problematic in several aspects*:

*a) To prepare mono-nucleosomes, the common nuclease of choice is MNase. However, the authors used DNase I, which efficiently cleave nucleosomal DNA at minor grooves, and Benzonase, which is even less proven enzyme to specifically digest linker DNA. More importantly, the nuclei DNA digestion pattern indicated a potential over-digestion (DNA fragment is smaller than 147bp), and there is no evidence that nucleosomes remain intact. This experiment should be repeated with MNase*.

*b) H2A, H2B and H4 western should be included in nucleosome pull-down when H3 antibodies are used*.

Following the reviewers’ suggestion we performed the co-immunoprecipitations with MNase-digested extracts. Our protocol yielded intact nucleosomes as revealed by the length of the DNA fragments (Figure 2—figure supplement 1) and by the western blot of the individual histones (Figure 2).

*c) In control cells (Left), when comparing IP to the input, K36me3 IP seems to put down K4me2 more efficiently than K36me3. This result would suggest that K4me2 co-occupies almost completely with K36me3, which is NOT consistent with numerous mass-spec studies and genome-wide analysis. The authors should elaborate on this result*.

To clarify the issue raised by the reviewers, we performed the H3K36me3/H3K4me2 co-immunoprecipitations on MNase-digested extracts. These new data (see Figure 10) now shows a very mild co-immunoprecipitation of the modified histones in control cells, which is consistent with the previous analysis mentioned by the reviewer. As we had reported in the original submission, H3K4me2 failed to co-purify with 53BP1. Since we think that the additional information provided by this result is very limited, we have decided to remove the H3K4me2 blot from the revised manuscript.Author response image 1.Co-immunoprecipitation of 53BP1 and H3K36me3 with H3K4me2. The experimental details are outlined in the manuscript. The H3K4me2 antibody was from Abcam (ab32356).

*6) To rule out indirect transcriptional effect caused by Setd2 knockdown, the authors settled to merely analyze the published RNA-seq data in Set2 depleted cells. However, this is not a correct analysis for this purpose. Gene expression comparison should have been done between WT and Setd2-KD cells under DNA damage induced conditions. This analysis should be done in WT and Setd2-KD cells under DNA damage induced conditions or removed from the manuscript entirely. If the authors decide to remove this section, they should acknowledge in the text the possibility of indirect SETD2 action via undefined transcriptional effects*.

Given the excessive amount of time that we need to perform de novo RNA-seq and data analysis, we have decided to remove the RNA-seq analysis from the manuscript. As requested by the reviewers, we acknowledge the possibility of indirect transcriptional effects caused by SETD2 knockdown (see Discussion).

7) The “Break & access model” proposed by authors in the Discussion section makes no sense to the reviewer for the following reasons:

*a) Recombinant Setd2 homologs in human,* Drosophila*, and yeast all show stronger activity toward nucleosomal substrates than histones; b) The works from the Briggs' lab (Du et al 2008 G&D and 2010 JBC) clearly showed that H2A surface is required for Set2-mediated K36me3; c) Based on the structure, the removal of H2A/H2B bears no influence on “disclose H3K36 substrates for Setd2-dependent trimethylation”. Please modify the model to more accurately depict the results obtained and the current state of knowledge*.

We thank the reviewer for this valuable input on our model. We now provide a more careful explanation of the “break and access model” taking into account the data referred to by the reviewer.

*Experiments using clear cell renal carcinoma cell lines involve only one of each wild type SETD2 versus mutant SETD2. Given that these cell lines are not isogenic, it is imperative to reproduce the key findings in*
Figures 5, 6 and 7
*in at least one other pair of wild type and mutant SET2D*.

We included new data obtained with another pair of ccRCC cells (*SETD2* wt and mutant) (Figure 6, Figure 8 and Figure 7—figure supplement 1). The results obtained support the original findings that *SETD2* mutant ccRCC cells have impaired DNA damage signaling, DNA repair and p53 checkpoint activation.